# Research Progress on the Synthetic Biology of Botanical Biopesticides

**DOI:** 10.3390/bioengineering9050207

**Published:** 2022-05-12

**Authors:** Jianbo Zhao, Dongmei Liang, Weiguo Li, Xiaoguang Yan, Jianjun Qiao, Qinggele Caiyin

**Affiliations:** Key Laboratory of Systems Bioengineering of Ministry of Education, School of Chemical Engineering and Technology, Tianjin University, Tianjin 300072, China; zhaojbooo@163.com (J.Z.); ldmxp@tju.edu.cn (D.L.); liweiguo099@126.com (W.L.); xgyoujian@126.com (X.Y.); jianjunq@tju.edu.cn (J.Q.)

**Keywords:** synthetic biology, botanical biopesticides, *emodin*, *celangulin*

## Abstract

The production and large-scale application of traditional chemical pesticides will bring environmental pollution and food safety problems. With the advantages of high safety and environmental friendliness, botanical biopesticides are in line with the development trend of modern agriculture and have gradually become the mainstream of modern pesticide development. However, the traditional production of botanical biopesticides has long been faced with prominent problems, such as limited source and supply, complicated production processes, and excessive consumption of resources. In recent years, the rapid development of synthetic biology will break through these bottlenecks, and many botanical biopesticides are produced using synthetic biology, such as *emodin*, *celangulin*, etc. This paper reviews the latest progress and application prospect of synthetic biology in the development of botanical pesticides so as to provide new ideas for the analysis of synthetic pathways and heterologous and efficient production of botanical biopesticides and accelerate the research process of synthetic biology of natural products.

## 1. Introduction

The widespread use of pesticides has become one of the most important factors in ensuring food production and supply [1], as pesticide use can recover 30% to 40% of the total crop losses worldwide [2]. Traditional chemical pesticides in production and application processes not only cause environmental pollution, such as the “three Rs” (residues, resistance, and re-rampant pests) [3], but also affect the quality and safety of agricultural products and endanger human life and health [4,5]. The attachment rate of chemical pesticides on crops was reported to be 10% to 30%, with the remaining portion entering the ecological cycle and introducing a certain amount of pollution into the ecological environment [6]. The presence of a large quantity of remaining chemical pesticides in the river will lead to water pollution, causing eutrophication of water bodies and further damaging the living environment of water-body organisms [7]. Nineteen pesticides were previously detected in seven typical watersheds in China (Yangtze River, Taihu Lake, Yellow River, Songhua River, Heilongjiang River, Grand Canal, and Dongjiang River), with an average concentration range of 0.02 to 332.75 ng/L [8]. The use of chemical pesticides for pest and disease control has increased resistance to chemical pesticides. Based on current reports, at least 16 kinds of diseases have developed resistance to 11 kinds of pesticides, including two species of rice, two species of wheat, one species of potato, and one species of vegetables; the resistance of wheat, potato, and vegetable diseases is the most serious [9]. The number of people poisoned by the use of chemical pesticides in the world is now up to more than 3 million each year, with more than 200,000 elderly people losing their lives [10]. Thus, the creation of novel sage and environmentally friendly pesticides become critical to ensure the safety of agricultural production [11].

There is no uniform and accurate definition of biopesticides globally. In 2017, the FAO/WHO jointly published the Guidelines for the Registration of Biopesticides of Microbial, botanical and Pheromone Classes for Plant Protection and Public Health, which means that biopesticides include microbial pesticides, botanical pesticides, and pheromone pesticides, while natural enemies, genetically modified organisms, and chemically synthesized substances with the same structure as natural compounds do not belong to this category [12]. In the United States, biopesticides refer to naturally existing substances that have pest control efficacy and function through non-toxicological mechanisms, including biochemical pesticides, microbial pesticides, and genetically modified protectants. It is worth noting that under the definition issued by the EPA, not all naturally occurring substances are biopesticides; for the sake of management, *pyrethrins*, *spinosad*, *avermectins*, etc., are still registered and managed according to chemical pesticides [13]. According to the Pesticide Registration Data Requirements implemented in China in 2017, biopesticides include three categories of biochemical pesticides, microbial pesticides, and botanical pesticides. Agricultural antibiotics are produced through microbial fermentation and therefore belong to the microbial pesticide category of biopesticides but are basically equivalent to chemical pesticides in terms of registration criteria and others [14]. It can be found that most countries and organizations generally recognize microbial pesticides and biochemical pesticides as biopesticides. FAO, the United Kingdom, Australia, China, and other countries and organizations recognize that pesticides of botanical origin also belong to biopesticides. The United States, Canada, and other countries recognize genetically modified insecticidal crops PIP as also belonging to biopesticides. In this paper, biopesticide refers to a formulation made from living organisms or their metabolites that has a killing or inhibiting effect on agricultural pests, including microbial-derived pesticides, botanical pesticides, natural enemy itself, animal-derived pesticides, antibiotics, biochemical pesticides, and plant-incorporated protectant (PIP) [15,16,17]. 

Compared with chemical pesticides, biopesticides have outstanding safety advantages. Firstly, biopesticides are more easily degradable. From a microscopic point of view, the ingredients of biopesticides are substances that exist in nature, which are easily decomposed and have a shorter safety interval, and do not cause major damage to the ecological balance. Secondly, biopesticides produce no or low residues. Sing et al. compared the effects of the application of the chemical pesticides *chlorpyrifos*, *cypermethrin*, and the biopesticide *azadirachtin* on the production of inter-root bacterial communities in *cowpea* and found that the residual concentrations of *chlorpyrifos* and *cypermethrin* in soil were 0.18 mg/kg and 0.86 mg/kg, respectively, at low doses and 0.36 mg/kg and 4.02 mg/kg, respectively, at high doses. While The residue levels of *azadirachtin* were not detected [18]. Up to now, there are no reports of residues or persistent toxicity of botanical biopesticides such as azadirachtin in the environment [19]. Thirdly, the toxicity of biopesticides is generally lower. For example, biogenic information compounds, synthesized artificially from plants, insects, etc., are released slowly into the field using a release device to interfere with the mating, feeding, and other behaviors of insects, thus reducing the population of target pests. This mechanism does not affect other organisms. Xu et al. compared the toxicity of *spinosad* and *cypermethrin*, and the results showed that the LC50 (24 h) of 25 g/L *spinosad* SC was 1.341 × 10^−5^ mg/cm^2^, with a safety factor of 0.054, and the LC50 (24 h) of 4.5% β-*cypermethrin* emulsion in water is 2.711 × 10^−5^ mg/cm^2^, with a safety factor of 0.120 [20]. There was also a study to measure the toxicity of *polyoxin*, *kasugamycin*, and *difenoconazole*, and the results showed that the EC50 values of *polyoxin* and *kasugamycin* were 60.184 ug·mL^−1^ and 86.438 ug·mL^−1^, respectively, which were significantly lower than that of *difenoconazole* 4.239 ug·mL^−1^ [21]. Fourthly, the application of biopesticides is difficult to produce resistance. Comparing the control effects of *matrine* AS and *bifenthrin* EC on tea looper (ectropis obliqua Prout), it was found that 3 d after application, the population reduction rate was 99.23% and 91.69%, respectively [22]. When the new biological miticide NBIF-001 wettable powder was used for 1 d, the adult mite body of Panonychus citri McGregor underwent lysis, and when the agent was used for 5 d, there were only insect residues on the leaves. In contrast, the Panonychus citri McGregor died after 5 d of using abamectin·spirodiclofen suspension concentrate, and the insects did not show cleavage, but the eggs increased, indicating that the Panonychus citri McGregor may have developed resistance to abamectin·spirodiclofen [23]. In addition, biopesticides have much fewer adverse effects on the soil. The diversity of soil microorganisms is the basis for maintaining healthy and stable agroecosystems. The effect of pesticides on soil microorganisms has become an important indicator for evaluating the ecological safety of pesticides in many countries. Take *Bacillus thuringiensis* (Bt) as an example, Bt protein will be undetectable in soils such as yellow-brown, red, and brick red soils for two weeks [24], and it also showed a gradual decrease in other soils [25]. There were no significant changes in bacteria and fungi in the soil where Cry1Ac protein had been applied within three months, and the diversity and homogeneity of microorganisms in the soil were not affected [26]. Furthermore, biopesticides have much less impact on other organisms in the ecosystem. For example, the field application of the Bt formulation XenTari did not affect the population of predatory and parasitic natural enemies [27]. Even directly feeding of Cry1Ab protein had no negative effect on the growth and development of chrysopa perla [28]. Feeding high doses of Cry1Ab/Ac protein to mice resulted in a rapid degradation in gastric and intestinal fluids, with no adverse effects on growth [29]. Volunteers received 1 g of Bt preparation (at a dose of 1011 CFU) for 5 d orally without adverse effects. At and below this dose, Bt and its products were not pathogenic or toxic to humans [30].

Biopesticides fit the development trends of modern agriculture and are gradually becoming mainstream in modern pesticide development. However, biopesticide production has long been restricted by prominent bottlenecks such as restricted resources and supplies, cumbersome production processes, and excessive resource consumption [31]. The rapid development of synthetic biotechnology in recent years has provided the possibility to break through these bottlenecks.

Synthetic biology belongs to an emerging field that aims to transform our ability to detect, manipulate, and interact with living systems by combining knowledge and techniques from biology, chemistry, computers, and other disciplines [32]. In the past two decades, synthetic biology has seen significant developments, including the design of minimal bacterial genomes by Hutchison et al. [33], the highly modified brewer’s yeast genome by Richardson et al. [34], the biosynthesis of the antimalarial drug precursor *artemisinic acid* [35], and the yeast metabolic engineering of opioids such as *tiobain* and *hydrocodone* [36]. With the continuous maturation of synthetic biotechnologies, such technologies have been widely used in biomedicine, industrial chemical products, and energy and agricultural products. In biomedicine, Ye et al. used the mitogen-activated protein kinase (MAPK) signaling pathway, which generates MAPK-mediated activation of transcription factors, to assemble an insulin-sensitive transcriptional control device that can precisely distinguish between physiologically normal and abnormally elevated blood insulin levels [37]. Using a lipid-sensing receptor (LSR) composed of the rhizome-response inhibitory factor TtgR and the peroxisome proliferator-activated receptor α (PPARα), Rossger et al. monitored fatty acid levels in the blood and applied pramlintide, a short appetite-suppressing peptide, to treat obesity [38]. In industry, Ahn et al. produced up to 134.25 g/L of *succinic acid* with a fractionated fermentation yield of 21.3 g/L-h by expressing malate dehydrogenase of *Corynebacterium glutamicum* origin in *Mannheimia succinogenes* [39]. In terms of energy, Huang et al. developed an integrated bioprocessing (CBP) system using recombinant brewer’s yeast to produce synergistic cellulase/amylase on the cell surface to overcome energy conversion limitations. After 72 h of fermentation, the *ethanol* production quality and yield reached 73 g/L and 64%, respectively, demonstrating a viable cell surface display fermentation system that can be used for the direct high-density production of *ethanol* from pigment-extracted microalgae material [40]. 

In recent years, synthetic biology has made it possible to design novel and specific metabolic pathways for desired chemical substances, offering unique advantages in the development and modification of biopesticides. In this study, we summarize the important applications of synthetic biotechnology in botanical pesticides and discuss this technology’s potential value and challenges in the present, along with outlooks for the future. Synthetic biology is advancing biology by expanding biomanufacturing capabilities and providing new insights into natural biological systems.

## 2. Botanical Biopesticides

### 2.1. Current Status of Botanical Pesticides Development

Botanical biopesticides are an important component of biopesticides. Such pesticides are a botanical formulation applied to the recipient plant according to a specific method based on stable active ingredients in the plant for protection against diseases, insects, and weeds. In general, such pesticides are generally contained in various types of substances such as alkaloids, glycosides, toxic proteins, volatile essential oils, tannins, resins, organic acids, esters, ketones, and terpenes. 

Over the past 20 years, the literature on pesticides based on botanical ingredients, such as insecticides, has increased from 2% to more than 21% of total publications, including literature and monographs [41]. Moreover, biopesticides are likely to grow from the current 4–5% of the global pesticide market to 20% by 2025. The growth of botanical pesticides is likely to be even stronger, rising from the current market share of 1–2% to more than 7% and growing at a rate of 16% per year [42]. Developed countries and regions such as Europe and the United States use 64% of the world’s botanical pesticides.

According to the World Food and Agriculture Organization statistics, the worldwide use of plant-derived biopesticides in 2022 is shown in Table 1. (“/” means that data are not available.)

At present, about 4000 varieties of plants with agricultural activities have been discovered in China, of which more than 2000 kinds have insecticidal activities and great potential for development [43]. As of 10 October 2020, the number of registrations of botanical pesticides in China reached 194 [43], mainly including alkaloids, terpenoids, organosulfur, esters, phenols, coumarins, anthraquinones, flavonoids, and isoflavonoids, specifically including *matrine*, *rotenone*, *azadirachtin*, *veratrine*, *pyrethrin*, *nicotine*, *celastrus angulatus*, *eucalyptol*, *camphor*, *dextrocamphor*, *anise oil*, *neochamaejasmin A*, *triptolide*, *curcumol*, *osthole*, *eugenol*, *physcion*, *cresol*, *berberine*, *sterol*, *saponin*, luowei, *bitter alkaloids*, *allicin*, *d-limonene*, *terpene alcohol*, *allyl isothiocyanate*, *ginkgo biloba* fruit extract (*pentadecenoic acid*, *tridecanoic acid*), and *psoralen corylifolia Linn.* seed extract (*isobavachalcone*) for a total of 28 plant species [44]. 

Plant-based pesticides do not guarantee safety, as raw-material acquisition, preparation processing, and the pesticide itself all have certain risks. Traditionally, the natural ingredients for *camphor*, an insecticide for hygiene, were extracted from the trunks of camphor trees, which once led to the excessive felling of these trees. Consequently, the natural sources of *camphor* available today are far from meeting the expanding market demands for *camphor* due to a lack of resources and conservation [45]. The intermediates and by-products produced during the production of pesticides are directly discharged into the environment with the “three wastes”, which puts pressure on the ecological environment [44]. The safety of botanical pesticides themselves also carries some risk. For example, *triptolide* is a rodenticide of plant origin, but from the viewpoint of the duration of toxic action, as well as the virulence of Rehmannia methylene, *triptolide* is actually an acute rodenticide and is not entirely safe. It is acutely toxic to some organisms other than rats and can easily cause secondary poisoning [44].

Table 2 shows the registration information of botanical pesticides in China.

### 2.2. Botanical Biopesticides and Synthetic Biology

Common botanical pesticides include the following. *Pyrethroids* come from *tanacetum cineraritfolium*, a plant of the pyrethrum family, and have a strong tactile action against a wide range of pests. It is a light yellow oily liquid that is insoluble in water. From a structural point of view, *pyrethroids* are condensed from an alcohol ligand and an acid ligand [46], and there are two types of acid ligands, chrysanthemic acid and pyrethric acid, and those containing chrysanthemic acid are type 1, and those containing pyrethric acid are type 2. There are three types of alcohol ligands: cinerolone, jasmolone, and pyrethrolone, and those containing different alcohol groups are named cinerin, jasmolin, and pyrethrin, respectively. *Azadirachtin* is mainly isolated from *AzadirachtaindicaA. Juss*, and is also found in *A. excelsa* [45], *A. siamensis* [46], and *Eupenicilliumparvum* [47]. This is a citrinin-like active substance belonging to the tetracyclic triterpenoids, which are structurally similar to hormonal substances such as steroids and steroidal organic compounds. *Matrine* is the first natural botanical insecticide species with clear composition and original production approval in China. Furthermore, it is also one of the products with the highest audience among botanical pesticides in China [46]. It is derived from the famous insecticide plant *Celastrusangulatus Maxim*, a vine of Celastrus in the family Celastraceae. Its insecticidal active ingredients are mainly polyol esters and their alkaloids with β-dihydro sedum furan as the framework [48]. *Carbamate* insecticides were created using *esserine* from *purple haricot* seeds as lead compounds, and *nicotine* insecticides were synthesized by modifying *nicotine* in *nicotiana tabacum* as the parent nucleus [43,44,45,46,47]. Their products are mainly plant extracts, such as *resorcin*, *reynoutriajaponicaHoutt*, *organum vulgare*, etc.

Currently, direct extraction from target plants or chemical synthesis is the main source of botanical biopesticides. In terms of direct extraction, some compounds are only present in certain parts of the target plants, such as *glycyrrhetinic acid* (only in the roots of *licorice*), *paclitaxel* (only produced in the bark of *bauhinia blakeana Dunn*) [49], *saponin* (extracted from the seeds of *Camellia sinensis*) [38], *isobavachalcone* (extracted from the source of the *psoralea corylifolia L.* seeds) [50,51], and *curcumol* (produced only in *Curcuma longa*) [52]. Therefore, direct extraction would be not only be wasteful to plant resources but also place great pressure on the environment due to a large number of organic reagents. In addition, natural products accumulate slowly and at low levels in the plant, and the use of artificial cultivation introduces problems such as long culture cycles, wasted land resources, and vulnerability to weather factors. Additionally, the fact that plants contain many structurally similar compounds creates great difficulties in isolation and increases costs [51]. These factors have led to difficulties in meeting research and market demands using traditional botanical production methods and an urgent need for an efficient and green sustainable method to natural plant natural products [53]. In terms of chemical synthesis, most natural products are characterized by complex structures and many chiral centers, which can lead to deactivated, difficult-to-separate, or even toxic spin isomers during the process of synthesis. At the same time, natural products require complex synthetic steps, low conversion rates, and high energy consumption. Additionally, the organic reagents used in synthetic processes also can put great pressure on the environment, making it difficult to meet the needs of industrialization [54]. 

The development of synthetic biology has prompted new ideas to solve the aforementioned problems. Synthetic biology is a discipline that genetically designs and modifies cells or living organisms by synthesizing biological functional components, devices, and systems to give them biological functions that meet human needs or even create new biological systems [55]. This discipline constructs cell factories or in vitro synthetic systems by systematically designing metabolic pathways and synthetic assemblies from a large number of genes, which can theoretically synthesize arbitrary compounds and will revolutionize the fields of drug manufacturing, industrial chemistry, and materials science. Biological systems are complex and unpredictable wholes. The continuous re-optimization of core molecular components through synthetic biotechniques and strategies can enhance the biological functions of existing systems, simulate and construct biological components that do not exist in nature, and create new biological functions and systems [56]. 

Plant natural products have long been a challenging and important area of study in synthetic biology research. Using microorganisms to synthesize structurally complex plant natural products can not only achieve continuous large-scale production but also obtain high purity and yield. At the same time, this process can obtain some structural analogues that are difficult to find in nature. Progress has been made in the heterologous synthesis of plant natural products in microbial cells using synthetic biology methods, especially in botanical pesticides. For example, the botanical biopesticide *celangulin*, a class of β-dihydroagarofuran sesquiterpene compounds, has a biosynthetic pathway that follows the common route of sesquiterpene compound synthesis: skeleton synthesis, cyclization, post-modification [57]. Firstly, IPP and DMAPP, the common precursors of all terpenes, are synthesized via the MVA pathway in the cytoplasm or MEP pathway in the limb. Secondly, two molecules of IPP and one molecule of DMAPP are catalyzed by farnesyl pyrophosphatases (FPPs) to synthesize FPP, which is then cyclized into the sesquiterpene skeleton via the action of sesquiterpene synthases. Finally, the structurally diverse sesquiterpenes are formed via the post-modification of P450 enzymes, BAHD acyltransferases [57]. Through transcriptome sequencing and differential gene expression analysis, our lab identified 15 sesquiterpene synthase genes, 8 P450 genes, and 4 BAHD acyltransferase genes that may be involved in the *celangulin* biosynthetic pathway. Through bioinformatic mining, a functional information dataset was established. The function of CaTPS2 was verified in vitro by recombinant protein, which was involved in the *celangulin* biosynthesis pathway and could catalyze the formation of γ-eudsemol [57]. Then, we integrated the CaTPS2 in the yeast chromosome multicopy locus, and the production of γ-eudsemol with an approximately 4.6-fold enhancement was obtained [57]. This laid the foundation for the development of synthetic biology in botanical pesticides. 

Currently, synthetic biotechniques are widely used in the synthesis of botanical biopesticides. In general, these techniques include the recombinant DNA technique, modular metabolic engineering, and genome editing. The recombinant DNA technique mainly includes the use of PCR and a Gibson assembly, which are widely employed in the synthesis of *pyrethroids* [58] *physcion* [59], and *eugenol* [60]. This technique can detect and analyze genes and targeted mutations. Genome editing includes zinc-finger nucleases (ZFNs), transcription activator-like effector nucleases (TALENs), clustered regulatory interspaced short palindromic repeats (CRISPR), and multiplex automated genome engineering (MAGE)/conjugative assembly genome engineering (CAGE) [61]. Table 3 shows the application of synthetic biotechniques in the biosynthesis of botanical biopesticides. 

Using *pyrethrin* biosynthesis as an example. Hu et al. overexpressed the gene encoding the synthase of *pyrethroid* acid ligands in *Chrysanthemum morifolium* and detected the release of volatile chrysanthemyl alcohol and chrysanthemyl-6-O-malonyl-β-D glucopyranoside accumulation, reaching 47 pmol/(h·g) of chrysanthemyl alcohol and 1.1 mmol/L of chrysanthemic acid [53]. Xu et al. reconstructed the chrysanthemic acid biosynthetic pathway in *Solanum lycopersicum*, overexpressing the TcCDS gene from *pyrethrum* and the ADH gene from wild tomato with ALDH, and reaching 67.1 μg/g FW of chrysanthemic acid, and 97% of the transferred DMAPP was converted [54]. Xu et al. reconstituted the synthetic pathway of pyrethric acid in a tobacco transient expression system, and the total yield of pyrethric acid reached (24.0 ± 2.7) μg/g [62].

Significant progress has also been made in the biosynthesis of rhodopsin. Tian et al. found that co-fermentation of *C. tigrinum* with a β-glucosidase producing *Rhizobium* strain T-34 was able to hydrolyze the glycosidic bond of rhodopsin β-glucoside in C. tigrinum, and after optimizing the fermentation and extraction conditions, the yield of *emodin* could be increased to 2.25% [50]. Lv found that *Aspergillus ochraceus* can produce large amounts of rhodopsin, and the content can reach 14.58 mg/mL by inducing high yield strain through ion injection and further optimizing the fermentation process [63].

There have also been major breakthroughs in the biosynthesis of some biopesticides. Cheng et al. designed an orthogonal *limonene* synthesis pathway, and a yield of 917 mg/L was achieved in yeast chassis and 1.3 g/L in *E. coli* through synthetic gene screening and the modification of the precursor metabolic pathway [64]. Sun et al. reconstructed the *rhodopsin* biosynthesis pathway in a *Saccharomyces cerevisiae* and achieved the highest yield reported in the literature by supplemental fermentation [65].

Synthetic biology research focuses on three levels: chassis, pathways, and components. Through the efforts of researchers, there have been many successful cases of the efficient synthesis of plant natural products in microbial chassis. The relevant research progress in recent years is briefly described below.

#### 2.2.1. Selection and Modification of Highly Adaptable Chassis 

The synthetic processes of botanical natural products are usually complex. To achieve their efficient synthesis in microbial chassis, the selection of the chassis is particularly important and should consider ease of handling, ease of cultivation, and adaptation to exogenous enzymes and products. *Saccharomyces cerevisiae* is a suitable chassis for the expression of natural products of plant origin. Using brewer’s yeast to express some pathways that require transmembrane proteins eliminates the need to modify transmembrane proteins. Moreover, these proteins contain microstructures such as mitochondria that mimic the subcellular localization of plant natural product synthesis [45]. For example, botanical P450 is a transmembrane protein that requires suitable membrane structures, such as the endoplasmic reticulum, for proper localization and folding. Nguyen et al. fused the key capsaicin gene sesquiterpene synthase EAS with cytochrome P450 EAH for *capsaicin* and expressed the gene in yeast (EPY300) [66]. Since acidification during fermentation alters the structure of the product, the heterologous expression of *capsaicin* was successfully achieved (250 mg/mL) after adding the buffer HEPES to adjust the pH to 7.0 during the configuration of the medium YPA [67]. Liu et al. heterologously expressed the oxidized squalene cyclase (OSC) genes AiOSC1 and MaOSC1 from *azadirachta indica* in the mevalonate pathway (MVA) optimized *S cerevisiae* and co-integrated AiOSC1 with AtAQS2 and PgtHMGR into the *delta* locus of the *S cerevisiae* chromosome to construct a recombinant strain with a high yield of the azadirachtin precursor substance tirucalla-7,24-dien-3β-ol, thereby providing a large number of substrates for the identification of important genes in the *azadirachtin* biosynthetic pathway [57]. 

Lipolytic yeast is another suitable chassis. Markham et al. applied four different metabolic engineering strategies using lipolytic yeast to increase the formation of an acetyl-coenzyme A precursor, an important precursor substance for the production of natural products such as terpenoids [68]. It was found that the enhancement of pyruvate branched pathways (pyruvate decarboxylase PDC, acetaldehyde dehydrogenase ALD, and acetyl-coenzyme A synthase ACS) and β-oxidation pathways yielded the most significant increase in cytoplasmic acetyl coenzyme A synthesis, while the synthesis of triacetate lactone (TAL) was promoted by the overexpression of the above pathways, resulting in a TAL yield of 35.9 g [68]. Liu et al. successfully achieved alternative pathways for acetyl coenzyme A synthesis, namely the acetate utilization pathway (acetyl coenzyme A carboxylase ACC1 and malic enzyme MAE1) and bacterial-derived cytoplasmic pyruvate dehydrogenase PDH in lipolytic yeast [69]. In addition, tobacco is an important model organism. Xu et al. explored the pathway for *pyrethrin* production in tobacco by co-expressing the key enzymes TcCDS, TcADH2, TcALDH1, TcCHH, and TcCCMT. The products were treated with glycoside extracts after fermentation, and the product *pyrethrin* 24 ± 2.7 μgg-1 was obtained via GC-MS analysis [62]. Trans pyrethrins are key substances for the synthesis of *pyrethrin*, and Xu et al. co-expressed the key enzymes TcCDS, TcADH2, and TcCDS, TcADH2, and TcALDH in tobacco to obtain 1328 and 818.4 nmol/g of trans pyrethrins, which were 48-fold and 122-fold higher, respectively, than TcCDS alone, indicating that the expression of *pyrethrin*, a key enzyme for *pyrethrin* synthesis, in tobacco can effectively increase pyrethrin production [59]. These studies yielded the highest yields ever achieved. It shows that botanical biopesticides are making breakthroughs and heralds their potential.

On the other hand, to achieve the efficient heterologous synthesis of natural products of plant origin, another effective strategy is to select strains that can produce high yields of synthetic precursors of target products as a chassis. Chassis bacteria with high production of both core metabolites and heterologous secondary metabolite intermediates are very valuable in synthetic biology studies. Chassis bacteria with high production of synthetic terpene precursors GPP/FPP or amino acids can provide the necessary raw materials for the microbial synthesis of thousands of natural products. Chassis bacteria that are capable of producing high quantities of certain secondary metabolic intermediates can then efficiently synthesize specific products; for example, chassis bacteria that can synthesize (*S*)-auroxinine ((*S*)-reticuline) can enable the microbial synthesis of a variety of benzylisoquinoline alkaloids (BIAs), including morphine [62] and narcotine [70]. An important advantage of microbial synthesis over chemical synthesis is the high yield of a series of intermediates required for secondary metabolic synthesis through metabolic pathway optimization. When such a chassis bacterium is successfully constructed, researchers can expand its downstream product spectrum and thus synthesize a series of related products. Hawkins and Smolke redesigned and optimized the first few steps of the biosynthetic pathway of BIAs by heterologously expressing cytochrome P450 hydroxylase (CYP80G2) and Corytuberine-N-methyltransferase in yeast, which together catalyzed the formation of Magnoflorine from (*S*)-reticuline [60], thereby completing the downstream step of BIA biosynthesis. De Loache et al. were the first to identify a dopamine-producing tyrosine hydroxylase (NMCH) in yeast in 2015, which led to the successful synthesis of (*S*)-reticulocyanine from tyrosine in *Saccharomyces cerevisiae* [71]. Considering De Loache’s study alongside Hawkins and Smolke’s downstream steps for the synthesis of BIAs biologically in *Saccharomyces cerevisiae* suggests that replacing plant extraction with the in vivo production of benzylisoquinoline alkaloids in microorganisms is possible.

After selecting a suitable expression system, the metabolic capacity of the chassis bacteria can be further optimized via gene knockout, the replacement of highly active metabolic genes, and the overexpression of exogenous metabolism-related genes. For example, Tu et al. successfully increased GGPP production by knocking out rox1, ypl062w, and yjl06w4, downregulating ERG9 and overexpressing genes such as tHMG1 and ERG20 [72]. Nakagawa et al. synthesized the basic backbone of BIAs from the simplest carbon sources (glycerol and glucose) in genetically modified Escherichia coli and synthesized an important branching point intermediate (S) in the biosynthetic pathway, reticulopanaxanthine. The established microbial fermentation system provided the possibility for the future low-cost synthesis of different BIAa in microorganisms [73]. Zhu et al. isolated two homologous MYC2 TF genes, TwMYC2a and TwMYC2b, from *Trichoderma reesei* hairy root and inhibited the synthesis of the precursor mithridate synthase genes TwTPS27a and TwTPS27b in the Y1H Gold yeast strain by binding to the E-box (CACATG) and T/G-box (CACGTT) motifs. The promoter activity of TwTPS27b resulted in a 65.38 DW μg/g level of tretinoin synthesis, a 1.67-fold increase compared to the control group [23]. Meadows et al. applied various techniques to reconfigure the core metabolism of yeast for high acetyl CoA production and confirmed that the modified yeast could enhance the production of the terpenoid farnesene by 25% by expressing genes related to the exogenous acetyl CoA involved in the synthesis pathway [74], which not only does not require more added sugar but also requires less oxygen, providing a good option for industrial fermentations with limited oxygen supply. *Rhodopsin* is a precursor substance of the botanical pesticide rhodopsin methyl ether, which is catalyzed by oxymethyltransferase to form rhodopsin methyl ether. Liu et al. used *Saccharomyces cerevisiae* BJ5464-NpgA as the chassis strain to synthesize rhodopsin by heterologously co-expressing the exogenous gene SlACAS-HyTE, the decarboxylase AFDC, and introducing the double mutant acetyl coenzyme A carboxylase ACC1S659A. S1157A was used to improve the yield of the precursor substance malonyl coenzyme A and thus the efficiency of *rhodopsin* synthesis, reorganize the rhodopsin synthesis pathway, and carry out supplemental fermentation of the engineered bacteria to produce 661.2 ± 50.5 mg/L *rhodopsin*, which enabled the heterologous synthesis of *rhodopsin* methyl ether [75]. This production implies a new breakthrough. 

#### 2.2.2. Analysis of Plant Natural Product Synthesis Pathways and Mining of Key Enzymes

Plant natural products are diverse, and most are structurally complex. Although the precursors of many products are derived from common primary metabolic processes, the post-modifications of these products are highly diverse and specific, and most of these synthetic genes do not exist in clusters such as microorganisms, which poses a great challenge to the elucidation of synthetic pathways and the mining of key enzymes. A more general mining tool is multi-omics association analysis. Two enzymes (a UDP-glycosyltransferase and a P450 oxidase) involved in the biosynthesis of *breviscapine* were identified via combined transcriptomic and genomic analysis, and a synthetic pathway was constructed in yeast [60]. 

Using the available genomes and transcriptomes of *azadirachta indica* and Brassicaceae species, Hodgson et al. identified an oxysqualene cyclase AiTTS1, which can produce triterpene backbone tirucalla-7,24-dien-3β-ol in the azadirachtin biosynthetic pathway. Co-expressed cytochrome P450 enzymes from *M. azedarach* (MaCYP71CD2 and MaCYP71BQ5) and *C. sinensis* (CsCYP71CD1 and CsCYP71BQ4) in *S cerevisiae*, the spontaneous hemiacetal ring was formated after three oxidations [76]. For example, in the biosynthesis of the isoquinoline alkaloid thebaine, the conversion of 7SOA ((7S)-salutaridinol-7-O-acetate) to thebaine can occur spontaneously by thebaine synthase (THS). Furthermore, since no protein catalyzing a similar reaction was previously reported, protein mass spectrometry (protein-MS) played a key role in the excavation of this enzyme [77]. Liang et al. experimentally cloned the EGS gene AhEGS of *E. coli*, constructed the prokaryotic expression vector pET28a-AhEGS-BL21, transformed *E. coli* BL21 (DE3) using recombinant pET28a-AhEGS, and produced a specific protein of about 35 kD in *E. coli*. Protein structural domain analysis showed that this product is a short-chain dehydrogenase/reductase dependent on coenzyme II (NADP), belongs to the PIP family, possesses typical characteristics of *eugenol* synthase, and is a key enzyme for the synthesis of *eugenol* [57].

In recent years, with the rapid development of DNA synthesis, sequencing, and analysis technologies, we have been able to resolve and express increasingly complex biosynthetic pathways. In 2018, scientists successfully completed the heterologous synthesis, in yeast, of noscapine, an anticancer compound consisting of 25 enzymes [78]. *Allicin* is a novel insecticide with a wide range of applications in agriculture. Li et al. completed the genome sequencing of *garlic* in 2020 [79] and screened the most likely several allicinases [80]; however, glutathione and methacrylic acid-binding key enzymes were not reported, and host selection for heterologous expression of *allicin* remains a challenge [81].

#### 2.2.3. Optimization and Integration of Synthetic Biology Components

Although the biosynthetic pathways of some plant natural products can be resolved by intensive mining and functional validation in vitro and in vivo, it is not easy to take a broadly well-defined pathway, functionally express it in a heterologous microbial host, and produce the target product [82,83]. Some enzymes cannot be expressed correctly or achieve their functions in heterologous hosts due to incorrect folding, post-transcriptional modifications, mislocalization, a lack of cofactors, non-optimal pH, non-natural substrates, product feedback inhibition, etc. Some of these problems can be solved or improved by means of enzyme engineering. 

Dai et al. cloned five GGPPS genes, key enzymes for terpene biosynthesis, from *ragweed* and introduced the cloned GGPPS and SmCPS/KSL genes into *E. coli*. By analyzing the expression of TwGGPPS in MeJA-induced suspension cells, the authors found that the accumulation of ragweed methicin was enhanced with an increase in TwGGPPS1 and TwGGPPS4 [84]. Kikuta et al. overexpressed and purified an enzyme homologous to GDSL lipase in *E. coli* and termed it TcGLIP. By expressing all the variants of TcGLIP, variants JN418993 and JN418996 were found to have similar acyltransferase activity, with JN418990 showing significantly higher activity. These findings enhanced our understanding of the structural basis of TcGLIP in exerting *pyrethrin* biosynthetic activity [85]. During attempts to synthesize *morphine* and its semisynthetic derivatives in yeast, researchers found that a considerable amount of the by-product neomorphine accumulates in the cells [85], mainly due to codeine ketone reductase (COR) activity, which can be isolated from the substrate after localizing COR to the endoplasmic reticulum using a tag, thereby allowing the spontaneous rearrangement of neomorphine to form codeine ketones. This method ultimately resulted in a seven-fold increase in the yield of the target product, *morphine*. Shao et al. used recombinant *E. coli* BL21 (DE3) p ET28a-agl A to efficiently express the glucosyltransferase gene agl A derived from *Xanthomonas campestris* and obtained the recombinant enzyme via isolation and purification, which catalyzed the glycosylation of *eugenol* with a product conversion rate of 75% and effectively increased the yield of *eugenol* synthase α-EG [86]. 

Morishige et al. further clarified the pathway of chimeric OMT in *berberine* biosynthesis by expressing recombinant chimeric methyltransferase in *E. coli*. The authors found that only the cell lysates of recombinant *E. coli* with pET-64′OMT possessed the methylation activity of demethylated dodecanol; i.e., the N-terminal end of 6-OMT and the C-terminal end of 4′-OMT were functionally complementary, thus allowing the chimeric OMT to play a role in the biosynthesis of *berberine* [87]. In *E. coli*, Sato expressed two recombinant methyltransferases, recombinant S-adenosyl L methionine scoulerine 9-O-methyltransferase and S-adenosyl-L-methionine. Coclaurine N methyltransferase, (S)-THC, and (R,S)-N-methyl tetrahydroberberine were prepared, and the accumulation of *berberine* was successfully induced via the introduction of the enzymes, suggesting that the relatively high ectopic expression of a novel branching enzyme is one of the key factors behind the diversification of *berberine* metabolism [88].

For many synthetic pathways in which enzymes are involved, some intermediate metabolites are not directly available. Consequently, the activity of some enzymes cannot be verified with certainty, which can also make the heterologous synthesis of plant natural products difficult. In this case, the longer synthetic pathways can be broken down into shorter modules, ideally with each module yielding the desired intermediate. Gao et al. introduced an optimal synthetic route into the diploid yeast strain YJ2X via modular engineering to achieve the integration of Sm CPS and Sm KSL, as well as the integration of BTS1 and ERG20, which significantly increased the production of mitiladib, the precursor of tretinoin, to 365 mg/L [89]. Zhao et al. successfully engineered the biosynthesis of (S)-scoulerine, (S)-tetrahydropalmatine, and (S)-corydalmine and their intermediates, ultimately increasing the yield of the *berberine* bridging enzyme, a key enzyme in the *berberine* biosynthetic pathway, to 1.19 mg/L [90]. This study provided a new direction for the biosynthesis of berberine [91] and has been successfully applied to the microbial synthesis of noscapine [92], sanguinarine [93], strictosidine [94], and lanosterin [65]. 

Another outstanding advantage of the microbial heterologous synthesis of plant natural products is that a series of derivatives can be obtained by combining various synthetic components and elements to potentially obtain products with better functionality. First, non-natural substrates can be added to a relatively well-defined pathway, and a verification of the substrate accommodation capacity of the enzyme can be performed using an in vitro or in vivo assay strategy. Valliere et al. developed a flexible cell-free isopentenylation system to detect the substrate specificity of isopentenyltransferases and successfully applied this method to the microbial production of *cannabis sativa L.* and their derivatives [95]. Second, a biocombinatorial synthesis strategy can be utilized. For example, the introduction of artemisia aldehyde Δ11 (13) double bond reductase 2 (AaDBR2) from *Artemisia annua* L. into the sesquiterpene lactone synthesis pathway of A. chrysanthemi produced new sesquiterpene lactones that had not been detected in a natural state, thereby providing new ideas for the development of sesquiterpene lactone structures with better properties [96]. The introduction of new functional enzymes in the native synthetic pathway represents another option to add modifications such as methylation, hydroxylation, and glycosylation to products and thereby generate new derivatives. For example, two halogenases were integrated into the medicinal plant *catharanthus roseus L.* to obtain catharanthine [97], and Cheng et al. broke the traditional CLB pathway of limonene production in brewer’s yeast through the heterologous expression of limonene synthase with geranyl diphosphate (GPP) as a substrate, starting from the synthesis of NPP from SlNDPS1 catalyzed by IPP and DMAPP (cis GPP). The authors also expressed *limonene* syn CSTS2 in *Saccharomyces cerevisiae* and selected the glucose-dependent promoter HXT1 to replace the endogenous ERG20 promoter. Finally, the authors performed shake flask fermentation with *ethanol* as a supplement and achieved a *limonene* yield of 917.7 mg/L; the highest yield reported to date [64].

## 3. Outlook

In the 17th century, nicotine in tobacco was discovered and used to control bruchuidae, unveiling the prelude to the commercialization of botanical pesticides. With the in-depth research on botanical active ingredients and the development of new extraction techniques such as ultrasonic extraction, microwave extraction, and supercritical fluid extraction, more and more biopesticides have been commercialized. According to globalnewswire statistics, the global biopesticides market was $4.3 billion in 2020 and is expected to grow at a CAGR of 14.7% over the next five years to reach $8.5 billion by 2025 [64]. This shows that there is a very broad market for biopesticides. Chemical Book counted the number of materials supplied for plant-derived biopesticides. It can be found that eugenol is one of the more commonly used botanical biopesticides, and its suppliers are located in 16 countries or international organizations. Among the suppliers of various botanical biopesticides in the United States, *eugenol* has the largest number of suppliers, reaching 95. Next is *d-Limonene*, with 70 suppliers producing and supplying the product. *Osthole* and *matrine* are involved in 46 suppliers each. In the UK, there are 13 manufacturers producing *eugeno*, *triptolide*, and *osthole*. Furthermore, 11 manufacturers produce *matrine*, *physcion*, and *eucalypto* [98]. In general, the US and the UK have more suppliers of botanical biopesticides among the various countries and international organizations. Correspondingly, these two countries also have larger production of botanical biopesticides. EPA statistics show that pyrethrin is the botanical biopesticide with the largest number of registrations in the United States, with about 30 related products. The rest of the related products are generally registered in the number 2–4, such as *azadirachtin*, *d-Limonene*, *cresol*, etc. There are also two *rotenone-*related products that are registered but restricted in use [99]. In recent years, the EU is also accelerating the development of botanical biopesticides. According to statistics, *pyrethrins*, *azadirachtin*, *Spinosad*, and many other botanical biopesticide products have received emergency authorization [100]. Furthermore, the development of botanical pesticides in China has grown rapidly. In 2012, the annual production of botanical pesticides in China exceeded the thousand-ton mark for *matrine* (5858t), *azadirachtin* (1554t), *neochamaejasmin A* (1170t). Moreover, *triptolide* (600t), *pyrethrin* (376t), *fennel oil* (354.5t), *rotenone* (287t), *osthole* (271t), *eugenol* (225t), and *camphor* (200t) all also reached more than 200t [101]. In China, for example, 75 new pesticide products were registered in 2017, including 22 biopesticides, and the top three products with the highest number of registrations were *matrine*, *rotenone*, and *azadirachtin*, accounting for about 80% of the total [101]. By the end of 2019, a total of 28 biopesticides were registered in China, involving 177 companies. Among them, there are more than 15 registered enterprises related to the production of the six major kinds of *matrine*, *pyrethrin*, *azadirachtin*, *osthole*, *rotenone*, and the hygienic insecticide camphor and 5 to 10 registered enterprises producing *eugenol*, *berberine*, *veratrine*, *celastrus angulatus*, *cresol*, and traditional *nicotine* [30]. The number of registered single doses of *matrine* bases accounts for 45% of the number of registered single doses of agricultural botanical pesticide varieties [30]. Meanwhile, the number of production lots of botanical pesticides in China has exceeded 3000, of which 94.6% of the production lots are for *pyrethrin* and their derivatives as active ingredients, while the remaining botanical pesticides and active ingredients are mainly involved in *azadirachtin*, *celastrus angulatus*, *matrine*, *rotenone*, *nicotine*, *camphor*, *eucalyptol*, *fennel oil*, *veratrine*, *physcion*, *osthole*, *eugenol*, and *cresol* [67]. It can be found that biopesticides are gaining popularity in developing countries represented by China.

Biopesticides with higher safety, more environmental friendliness, and other advantages, in line with the developmental trends of modern agriculture, are gradually becoming mainstream in modern pesticide development. However, at present, natural biological pesticide types are still relatively unique and expensive with slow efficacy globally, and the development of new varieties and technologies is not yet comprehensive and rapid. The production of biopesticides has long suffered from bottlenecks, problems such as backward manufacturing processes, long production cycles, and an over-dependence on natural resources. Developing new compounds with novel structures and achieving the efficient production of known structures require the help of synthetic biology. Therefore, it is imperative to promote the application of synthetic biology in the field of biopesticide development for the development of the biopesticide industry.

There are three main methods for the production of biopesticides: direct extraction from organisms, chemical synthesis, and biosynthesis. Common extraction methods include extraction and separation using organic solvents such as ethanol, extraction by supercritical fluid extraction (SFE), and ultrasonic cycle technology, which means adding ultrasonic cycle processing to organic solvent extraction. These methods can reduce the number of organic solvents used, shorten the working time, and improve the extraction rate. With the development of science and technology, separation techniques such as chromatography have also provided new ideas for the extraction of natural products [102]. However, the direct extraction method relies entirely on the plant itself, which grows slowly, has a long production cycle for natural products, and has a limited yield, making it difficult to carry out large-scale production. There are two main disadvantages of chemical synthesis. Firstly, many natural products are structurally complex and contain many chiral centers, thus, deactivated and difficult to separate spin isomers can occur during the process of performing synthesis. Secondly, the synthesis has more steps, low conversion rate, high energy consumption, and the organic reagents used can put serious pressure on the environment, making it difficult to meet the needs of industrialization [103]. The introduction of synthetic biotechnology into the development of botanical biopesticides can effectively solve the problems of limited resources, complex production processes, and low production efficiency faced by traditional production, thus reducing the production costs of biopesticides and lowering the price barriers to their application. These measures would make biopesticides truly affordable for farmers in China and effectively reduce the number of chemical pesticides applied for the benefit of the environment. 

Synthetic biology has a wide scope for development. In 2006, the biosynthesis of *artemisinic acid* in *Saccharomyces cerevisiae* was first reported in *Nature* [35]. In 2013, by introducing plant dehydrogenase and cytochrome CYB5, regulating gene expression, and using a two-phase fermentation method, the yield of *artemisinic acid* was finally increased to 25 g/L^−1^ [104], laying the foundation for its industrial production and also signals a promising future for synthetic biology. Breakthroughs have been made in the biosynthesis and synthetic biology of plant-derived biopesticides, but there are still many pressing issues that need to be addressed. For example, the biosynthetic pathways are not fully resolved, and the coordination and balance between components need to be overcome after multiple components are assembled together. The future of synthetic biology remains focused on three levels: chassis, pathways, and components. Firstly, select and modify a highly adaptable chassis. In addition to microbial systems, plant systems are also functional gene heterologous expression systems that have been applied more often in recent years [105,106]. Plants themselves contain or encode enzyme genes, organelles with similar structure and function, coenzymes, coenzyme factors and precursors, which facilitate the expression and post-translational modification of botanical proteins. For example, Lycopersicon esculentum has been widely used for the production of natural products such as *astaxanthin* [107], *cyanidin* [108], and *flavonols* [109]. *Nicotiana benthamiana* has been used as a chassis to reconstruct the pathways of several natural products, such as *vinblastine* [110] and *podophyllotoxin* [111]. In the future, we can continue to explore more transient expression systems and try to realize the expression of botanical biopesticides in plant systems. At the same time, the modification of common chassis cells such as *Saccharomyces cerevisiae* and *E. coli* was performed. The expression and functional studies of P450 in *E. coli* were achieved by modifying the membrane localization sequence of the P450 gene [112]. A cytochrome P450 reductase from *Arabidopsis thaliana* was integrated on the *Saccharomyces cerevisiae* chromosome to construct a strain WAT11 dedicated to plant P450 expression and functional studies [113]. This strain has been widely used for P450 gene expression and functional studies. In the future, more suitable chassis need to be discovered, and the metabolic capacity of the chassis strain needs to be further optimized. Secondly, optimize the synthesis pathway. The biosensor plays an important role. With the specific sensitivity to the target metabolites, the information of metabolite changes can be outputted in real-time by various signals, which can realize dynamic monitoring and feedback to balance the biosynthetic pathway and improve the yield of natural products [114]. Chou et al. [115] constructed a sensor-actuator circuit for the dynamic control of genome-wide mutation rate, which successfully led to a five-fold and three-fold increase in the yield of *tyrosine* and *lycopen*e. It is believed that the future development of biosensors will further optimize the synthesis pathway and effectively contribute to the research on yield enhancement of natural product genetically engineered bacteria and explore more ways to mine components. Thirdly, the application of computational biology methods can obtain candidate genes and perform functional studies more precisely, effectively reducing workload and improving screening efficiency. The researchers used computational biology to mine the database for possible functional components of natural product biosynthetic pathways and realized the biosynthesis of thebaine and hydrocodone [36], isoquinoline alkaloids [116], and other natural products through functional studies and pathway reconstruction. 

It is believed that, with plenty of plants and microbial resources as the foundation, and synthetic biology development as the means, the quick advances in the study of botanical biopesticides will be promising. The development of varieties and efficient manufacturing will go hand in hand, driving the development of the entire biopesticide industry and providing powerful support for the healthy and rapid development of green agriculture.

## Figures and Tables

**Table 1 bioengineering-09-00207-t001:** Botanical pesticides used worldwide in 2022.

Area	Capacity (Insecticide—Pyrethroids) (Tonnes)	Capacity (Insecticide—Botanical Products and Biologicals) (Tonnes)	Capacity (Seed Treat Fung—Botanical Products and Biologicals) (Tonnes)
Japan	156	/	/
Armenia	15	1	/
Austria	20	6	/
Belgium	14	3	/
China, Hong Kong	1	2	/
Croatia	10	0	/
Cyprus	6	3	/
French Polynesia	1	1	/
Germany	113	25	/
Iceland	0	0	0
Italy	174	60	/
Lithuania	21	/	/
Madagascar	47	1	0
Malaysia	648	101	0
Maldives	64	14	/
Myanmar	195	36	/
New Caledonia	1	0	/
Panama	12	0	/
Paraguay	975	50	0
Poland	98	26	0
Saint Kitts and Nevis	1	0	/
Saudi Arabia	4732	2631	0
Slovakia	19	3	0
Slovenia	1	0	/
Sudan	224	0	0
Suriname	44	2	/
Switzerland	3	6	/
Togo	270	12	/
Ukraine	271	0	/
UK	28	10	0

Data Source: Food and Agriculture Organization of the United Nations, FAO, https://www.fao.org/faostat/en/#data/RP (accessed on 10 December 2021).

**Table 2 bioengineering-09-00207-t002:** Registration information of botanical pesticides in China.

Varieties	Purpose	Category	Formulation (Including TC)/Main Formulation	Registration Number
*Nicotine*	Insecticide	alkaloid	2/aqueous solutions, AS	8
*Matrine*	Bactericide	alkaloid	5/aqueous solutions, AS	122
*Berberine*	Insecticide	alkaloid	5/aqueous solutions, AS	8
*Veratrine*	Insecticide	alkaloid	2/aqueous solutions, AS	8
*Azadirachtin*	Insecticide	Terpenoids	1/soluble concentrate, SL	26
*Celastrus angulatus*	Insecticide	Terpenoids	6/emulsifiable concentrate, EC	7
*Triptolide*	Rodenticide	Terpenoids	3/emulsion in water, EW	2
*Curcumol*	Rodenticide	Terpenoids	2/Granules	2
*d-Limonene*	Insecticide	Terpenoids	2/Bait, RB	2
*Eucalyptol*	Insecticide	Terpenoids		2
*Cresol*	Bactericide	Terpenoids	2/soluble concentrate, SL	7
Luowei	Insecticide	Terpenoids	3/aqueous solutions, AS	2
*Saponin*	Insecticide	Terpenoid saponins	2/dust powder, DP	1
*Allicin*	Bactericide	Organic sulfur	1/aqueous solutions, AS	36
*Pyrethrin*	Insecticide	Esters	2/microemulsion, ME	27
*Eugenol*	Bactericide	Phenols	5/emulsion in water, EW	9
*Osthole*	Bactericide	Coumarins	2/soluble concentrate, SL	21
*physcion*	Bactericide	Anthraquinones	6/emulsion in water, EW	6
*Neochamaejasmin A*	Insecticide	Flavonoids	4/aqueous solutions, AS	2
*Isobavachalcone*	Bactericide	Flavonoids	2/emulsion in water, EW	2
*Rotenone*	Insecticide	Isoflavones	2/microemulsion, ME	23

Data Source: Ministry of Agriculture and Rural Affairs of the People’s Republic of China, China Pesticides Information Network, http://www.chinapesticide.org.cn/hysj/index.jhtml (accessed on 18 February 2022).

**Table 3 bioengineering-09-00207-t003:** Synthetic biotechniques in the biosynthesis of botanical biopesticides.

Item	Purpose	Technique	Application
Recombinant DNA Technique	Gene testing and analysis; Site-directed mutagenesis	PCR	Eg. Limonene biosynthesis: synthetic gene screening [64]
		Gibson assembly	
Modular Metabolic Engineering	Optimize metabolic pathways	Modular metabolic engineering	Eg. Pyrethrin biosynthesis: Reconstructed the chrysanthemic acid biosynthetic pathway in tomato fruit [62]; Rhodopsin biosynthesis: co-fermentation of C. tigrinum with strain T-34, optimizing the fermentation and extraction conditions [50]
Genome Editing	Edit Genes	Zinc-finger nuclease, ZFN	Eg. Pyrethrin biosynthesis: overexpressed the gene encoding the synthase of pyrethroid acid ligands in Chrysanthemum morifolium [52]
		Transcription activator-like effector nuclease, TALEN	
		Clustered regulatory interspaced short palindromic repeat, CRISPR	
		multiplex automated genome engineering, MAGE/conjugative assembly genome engineering, CAGE	

## Data Availability

Not applicable.

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
