# Peer review of "Research Progress on the Synthetic Biology of Botanical Biopesticides"

_bioengineering, 2022, doi:10.3390/bioengineering9050207_

Round 1
Reviewer 1 Report
Dear authors,
I made comments on the manuscript pdf.
The most important ones are:
- The authors assume biopesticides are safe but I don't see any convincing evidence. My impression is that the environmental risks have not been rigorously investigated. If carefully investigated, many risks may be detected.
- The authors state that there is no clear definition of biopesticides, but I think this ambiguity makes the manuscript confusing. The authors should provide definition of biopesticide in the paper.
- What is the current stage of synthetic biology of botanical biopesticides in terms of product development? How close is it for mass production and commercialization?
I hope my comments are of help.
Sincerely.

Reviewer 2 Report
- The manuscript is very comprehensive and the authors have done an excellent job with the organization.
- The language is concise and easy to read.
Author Response
Dear Reviewer 3,
Thanks very much for your time on our manuscript “Research Progress on the Synthetic Biology of Botanical Biopesticides”.
All of the valuable comments and corrections provided by the editor and reviewers have been taken into full consideration in the revising of the manuscript, and all of the changes made upon the original manuscript have been highlighted in red in the copy of revised manuscript. We hope the revised manuscript could meet the high publication standards of Bioengineering.
Thanks again for your help and please let us know if you have any question. With best regards!
Sincerely yours,
Qinggele Caiyin, Ph.D Professor
-------------------------------------------------------------------------------
Qinggele Caiyin, Ph.D Professor
School of Chemical Engineering & Technology
TianjinUniversity, Tianjin 300072
E-mail: qinggele@tju.edu.cn
Reviewer 3 Report
- Provide the details of Botanical Biopesticides in “2.2.”, such as chemical structure, basic properties and synthetic biotechniques, et al.
- Add the application examples (pyrethroids, physcion, and eugenol,et al) of synthetic biotechniques in Table 3 (Line 260 ).
- Correct“Haiyang Xu et al.” as “Xu et al.”(Line 299); Correct “Cheng S. et al.” as “Cheng et al.”(Line 471).
Round 2
Reviewer 1 Report
Dear authors,
Thank you for your revision. I think the ms has been well improved during a short period. However, I still found some points to be amended.
First, I suggest you focus more on the synthetic biology in the last Outlook section. Summarize the current progress of synthetic biology of botanical biopesticides and its issues to be addressed in the future. To me it seems that the present ms just mentions that more botanical biopesticides will be used in China in the future.
Second, there are so many scientific names that should be italicized. I commented this in the previous review but many of them are not yet fixed. Please do check the entire ms very carefully.
I made comments on some other specific points in the pdf.
Sincerely.
